# Spatial distribution of argan-tree influence on soil properties in South Morocco

Mario Kirchhoff[1], Tobias Romes[1], Irene Marzolff[2], Manuel Seeger[1], Ali Aït Hssaine[3], Johannes B. Ries[1]

[1]Department of Physical Geography, Trier University, Trier, 54286, Germany
[2]Department of Physical Geography, Goethe University Frankfurt am Main, Frankfurt am Main, 60438, Germany
[3]Department of Geography, Université Ibn Zohr, Agadir, 80060, Morocco

*Correspondence to*: Mario Kirchhoff (kirchhoff@uni-trier.de)

**Abstract.** The endemic argan tree (*Argania spinosa*) populations in South Morocco are highly degraded due to overbrowsing, illegal firewood extraction and the expansion of intensive agriculture. Bare areas between the isolated trees increase due to
limited regrowth, however, it is unknown if the trees influence the soil of the intertree areas. Hypothetically, spatial differences of soil parameters of the intertree area should result from translocation of litter or soil particles (by runoff and erosion or wind drift) from canopy-covered areas to the intertree areas. 385 soil samples were taken around the tree from the trunk along the tree drip line (within and outside the tree area) as well as the intertree area between two trees in four directions (upslope, downslope and in both directions parallel to the slope) up to 50 m distance from the tree. They were analysed for gravimetric
soil water content, pH, electrical conductivity, percolation stability, total nitrogen content (TN), content of soil organic carbon (SOC) and C/N ratio. 74 tension-disc infiltrometer experiments were performed near the tree drip line, within and outside the tree area, to measure the unsaturated hydraulic conductivity. We found that the tree influence on its surrounding intertree area is limited, with e.g., SOC- and TN-content decreasing significantly from tree trunk (SOC: 4.4 %, TN: 0.3 %) to tree drip line (SOC: 2.0 %, TN: 0.2 %). However, intertree areas near the tree drip line (SOC: 1.3 %, TN: 0.2 %) differed significantly from
intertree areas between two trees (SOC: 1.0 %, TN: 0.1 %), yet only with a small effect. Trends for spatial patterns could be found in eastern and downslope directions due to wind drift and slope wash. Soil water content was highest in the north due to shade from the midday sun, the influence extended to the intertree areas. The unsaturated hydraulic conductivity also showed significant differences between areas within and outside the tree area near the tree drip line. This was the case on sites under different land usages (silvopastoral, agricultural), slope gradients or tree densities. Although only limited influence of the tree
on its intertree area was found, the spatial pattern around the tree suggests that reforestation measures should be aimed around tree shelters in northern or eastern directions with higher soil water content, TN- or SOC-content to ensure seedling survival, along with measures to prevent overgrazing.

## 1 Introduction

The degradation of dryland forests is a major problem, since trees help prevent erosion and desertification (Dregne, 2002;
FAO, 2019; Verón et al., 2006). Due to a sparser vegetation cover, lower amounts of organic matter and rare but intense rain

events, soils in those regions are generally more vulnerable to erosive processes, which are result and cause for degradation and desertification (Ravi et al., 2010; Vásquez-Méndez et al., 2011).

Vegetation commonly affects the underlying soil and modifies the characteristics of soil, e.g., making it more resistant to erosive processes (Ludwig et al., 2005; Stocking and Elwell, 1976; Zhou et al., 2008). The impact on soils caused by vegetation becomes noticeable in terms of island structures, where higher concentrations of nutrients occur as a result of a strong accumulation under the protective cover of the canopy, especially in areas showing higher rates of drought (Allington and Valone, 2013; Garner and Steinberger, 1989; Ridolfi et al., 2008; Schlesinger et al., 1990). The higher amount of nutrients, especially the higher input of organic matter in soils underneath trees leads to a higher resilience against erosion and is responsible for the darker insular appearance (Auerswald, 1995; de Boever et al., 2015; Pérez, 2019). Besides the added fertility to the soil, dryland forests are also an important source of fodder for grazing livestock, making silvopastoral systems the most characteristic form of land use in drylands (Solorio et al., 2017; Soni et al., 2016).

The endemic woodlands of *Argania spinosa* are an example for an agro-silvopastoral system, which covers an area of 950.000 ha and is mainly located around the Souss basin in southwestern Morocco. It is well adapted to the high temperatures and water scarcity of South Morocco (Defaa et al., 2015; Ehrig, 1974; Mensching, 1957). The argan forest differs from other silvopastoral systems due to its complex usage rights involving grazing and browsing of the trees by local and nomadic herds of goats, sheep and camels as well as rainfed cultivation of cereals between the trees (highly speculative due to high variation in precipitation) and collection of the fruits to harvest the valuable cosmetic or alimentary argan oil (Alados and El Aich, 2008; Charrouf and Guillaume, 2009; le Polain de Waroux and Lambin, 2012; Lybbert et al., 2010). In the past, the high fuel value of argan wood resulted in the deforestation of the argan forest for use in the sugarcane industry or for sale as charcoal (Aït Hssaine, 2002; Faouzi, 2013). Although the argan woodlands have been designated a UNESCO biosphere reserve (Charrouf and Guillaume, 2018), afforestation programmes are ongoing and some felled trees have resprouted, the slow growth of *Argania spinosa*, the high grazing pressure and ongoing illegal firewood harvesting have led to a degradation of the woodlands that is visible in the architecture or growth form (Culmsee, 2005; Kirchhoff et al., 2019a; Marzolff et al., 2020) and the decrease of forest density (up to 44.5% decline of wooded area between 1970-2007) (le Polain de Waroux and Lambin, 2012).

This change in forest density has serious consequences on the soil. Previous tests carried out on several sites of *Argania spinosa* already confirmed higher nutrient and lower soil erodibility levels under the canopy compared to bare intertree areas, especially due to tree litter, higher soil water content due to shade and the stem acting as an obstacle against erosion. These degraded intertree areas do not provide a good basis for young sprouts to develop. With regrowth hindered and old trees being cut or dying-off the decline of forest density will only increase (Kirchhoff et al., 2019a). However, wind drift or slope runoff may move litter and soil material into the intertree area, thus possibly affecting soil parameters like soil organic carbon or total nitrogen content outside of the area covered by canopy (Pérez, 2019). The tree also provides shade for a part of the intertree area, which moves with the sun around the tree. The shade should have an effect on soil water content, hypothetically with the highest soil water content from the north of the tree (when the sun reaches its zenith in the south) to the east of the tree (shadows

grow longer, sun is lowering in the west, but air temperatures have increased compared to the morning) due to limited

evaporation (see 2.2 Experimental design).

This hypothetical spatial pattern of influence of the tree on the soil has only been researched for pine trees (Zinke, 1962) but not at all for argan trees or dryland forests, whereas the differences between tree or shrub vegetation and their corresponding intertree/intershrub areas have been investigated before, especially in 'fertile island'-research (e.g., Belsky et al., 1993; Boettcher and Kalisz, 1990; de Boever et al., 2015; Pérez, 2019; Qu et al., 2018). In South Morocco, where the geomorphologic

processes are highly dynamic (Aït Hssaine, 2002; Kirchhoff et al., 2019b; Marzen et al., 2020; Peter et al., 2014), it is likely that litter and soil particles are dislocated to the intertree areas. The knowledge about this possible dislocation and improvement of soil parameter values in the intertree areas could enable a better regrowth in these areas (Boulmane et al., 2017; Defaa et al., 2015) or show the need for rehabilitation by limiting degradation factors like overgrazing.

The aim of this study is therefore to analyse the spatial distribution of the influences an individual argan tree has on soil

properties of the intertree areas. For this purpose, we define

- "tree area" as the area covered by canopy (within the tree drip line),
- "intertree area" as all area not covered by canopy (i.e., between tree areas).

## 2 Material and methods

### 2.1 Study areas

The three study areas Ida-Outanane, Taroudant and Aït Baha are located in the western part of the Souss basin (Fig. 1, between 30° and 31° northern latitude and 9° and 7° western longitude). 30 test sites (one ha each) were chosen in these three environmentally differing study areas in order to cover varying altitudes, climate conditions and soil types (see Kirchhoff et al., 2019a).

Ida-Outanane is located on the southern foothills of the High Atlas close to Agadir and the Atlantic Ocean. Thus, its climate

is more maritime with temperatures very rarely exceeding 30 °C and precipitation ranging from 230-260 mm (data for the suburbs of Agadir, 20 km away) (Díaz-Barradas et al., 2010; Saidi, 1995). Soils are mostly immature with Regosols, Leptosols and Fluvisols (Jones et al., 2013) covering Paleozoic, Mesozoic and Cenozoic rocks of the High Atlas (Hssaisoune et al., 2016). Traditional rainfed agriculture (mostly wheat cultivation) is practiced on three out of six test sites between the argan trees while the rest is under silvopastoral land use.

The study area of Taroudant also lies in the southern foothills of the High Atlas, but is situated further inland about 80 km from the coast. The climate is more continental with 220 mm annual precipitation and a mean annual temperature of 20 °C (Peter et al., 2014; Saidi, 1995). Eleven test sites are situated in the study area, seven on a loamy alluvial fan covering the Pliocene and Quaternary fluvial, fluvio-lacustrine and aeolian deposits of the Souss basin (Aït Hssaine and Bridgland, 2009; Chakir et al., 2014), the other four sites are located on the foothills of the High Atlas. The vegetation is mainly characterized

by *Argania spinosa* as well as other shrubs and bushes such as *Launaea arborescens*, *Ziziphus lotus*, *Acacia gummifera*,

*Euphorbia* spec. and *Artemisia* spec. (Ain-Lhout et al., 2016; Peter et al., 2014; Zunzunegui et al., 2017). However, a dynamic land use change has been taken place in the Souss basin for several decades, with traditional rainfed agriculture and argan trees being replaced by more profitable irrigated citrus plantations as well as greenhouses for banana and vegetable cultivation (d'Oleire-Oltmanns et al., 2012; Kirchhoff et al., 2019b; Peter et al., 2014).

The study area Aït Baha is located on the northern foothills of the Anti-Atlas Mountains. Precipitation ranges from 250-350 mm annually and the annual temperature averages 18.7 °C (Seif-Ennasr et al., 2016). The Anti-Atlas is mostly made up of Precambrian and Paleozoic rocks, which are covered by Fluvisols (Jones et al., 2013) as well as Regosols and Leptosols. Thirteen test sites are situated in this study area, with three on argan reforestation sites that often yield mixed results (Defaa et al., 2015). Silvopastoral land use dominates on most test sites with cereals being cultivated between the argan trees on 105    ploughing terraces (on three test sites).

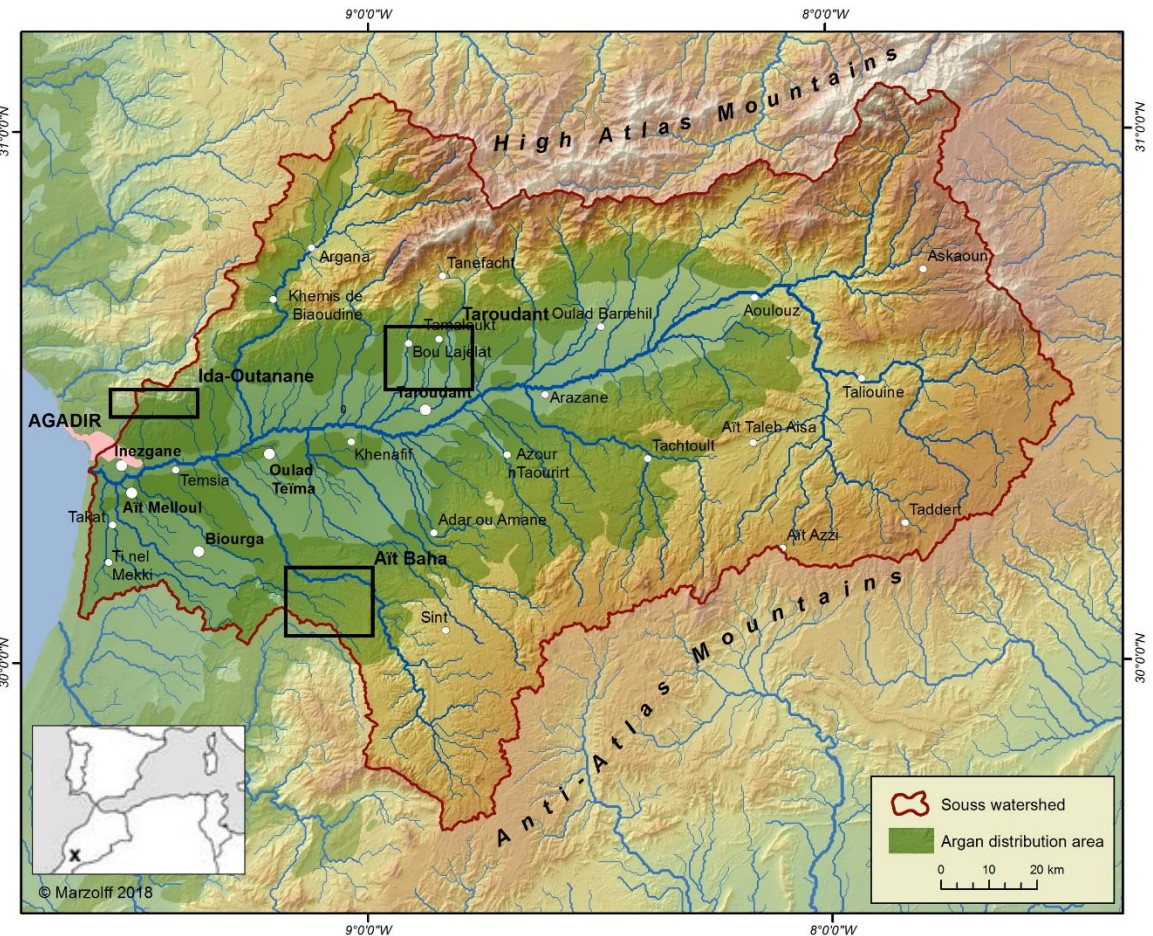

**Figure 1: Study areas Aït Baha, Ida-Outanane and Taroudant in the Souss basin shown in black rectangles. Argan distribution area shown in green shades (map modified from d'Oleire-Oltmanns et al., 2012).**

## 2.2 Experimental design

The potential influences of the tree on the intertree area are shown in Fig. 2 (in eastern directions due to wind drift, downslope due to hillslope wash, north due to shade in the midday sun). One tree per test site was investigated, except on one test site where sampling was undertaken around two trees (31 trees in total), therefore sampling 13 trees in the study area of Aït Baha,

seven trees in the study area Ida-Outanane and eleven trees in the study area of Taroudant. The trees were chosen to be as representative as possible for their test site, with regard to their size, degradation status and the distance between a sampling tree and its neighbour (see Kirchhoff et al., 2019a). Therefore, sampled trees were between 1 and 8 m high and varied from tall trees with round crowns to very dense shrub-like tree forms. Tree density varied from 3 to 292 trees ha$^{-1}$. In most test sites, the tree areas showed a higher vegetation cover than the intertree areas while the intertree areas were bare with a medium to

high stone cover.

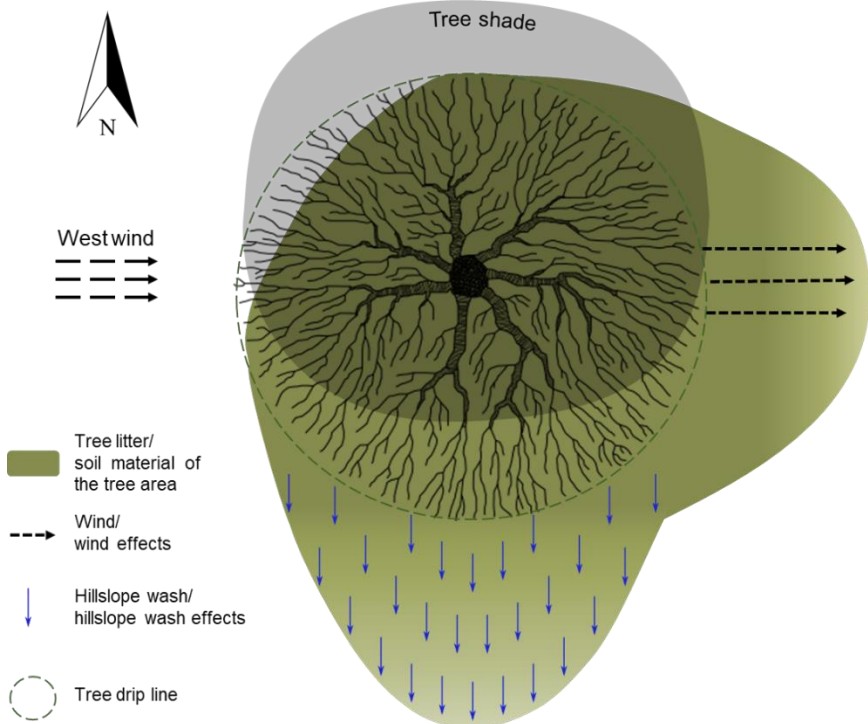

**Figure 2: Potential spatial pattern of tree influence on the intertree area.**

To measure the potential influence of the tree on the intertree area shown in Fig. 2, we took 13 soil samples around each tree.

One was taken next to the trunk (T1), while we took three soil samples each in four directions around the tree, namely upslope, downslope and in both directions parallel to the contour lines. The three samples in each direction were taken with increasing

distance from the tree trunk, one near the tree drip line under the canopy (T2), the next near the tree drip line just outside the crown's cover (IT3), and the third in the intertree area at the midpoint between the tree and its next neighbouring tree in that direction (IT4, Fig. 3). The two samples at the tree drip line (T2 & IT3) were generally about one metre apart, depending on
the crown's shape and the surface conditions.

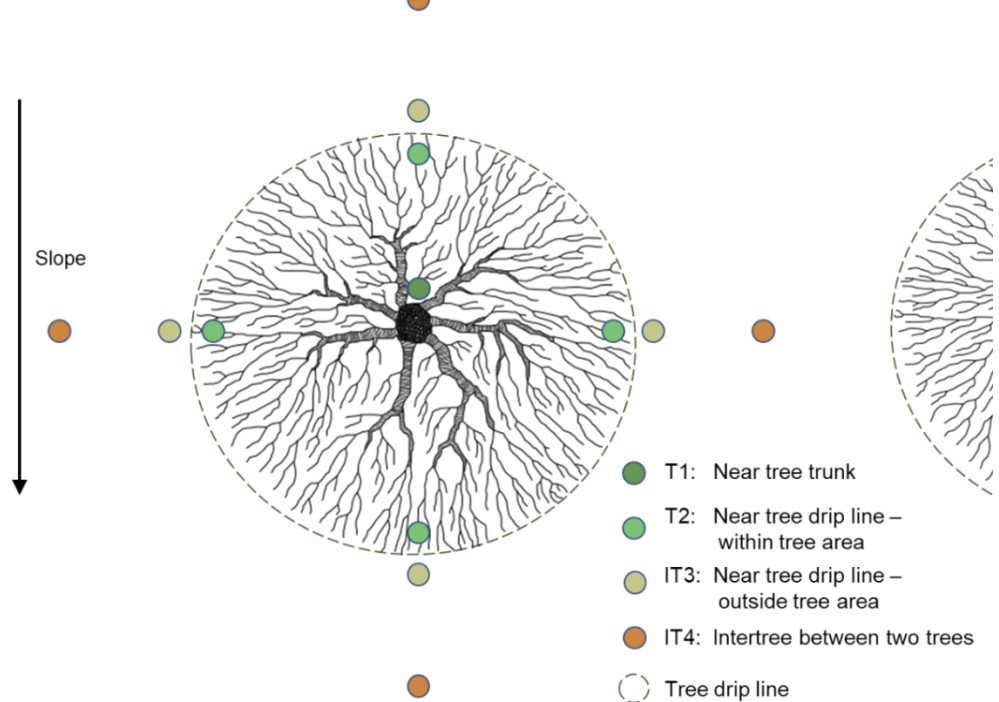

**Figure 3: Schematic diagram of soil sampling locations in the tree and intertree area.**

The 385 disturbed surface soil samples were taken during a field campaign in February and March 2019 up to a depth of 5 cm.
Since not all slopes were south-facing, transect directions were recorded in 8 directions of 45° angles each (e.g., N = 337.5 – 22.5°, NE = 22.5 – 67.5°). As the argan forest does not grow in a perfect grid pattern and the nearest tree in the sampling direction was not always in the exact direction needed, 90° differences between transect directions could not always be assured, but could vary some degrees to the left or right. For some trees less soil samples were taken due to the different tree densities and different tree architectures. For very dense, shrub-like trees it sometimes was not possible to sample the soil at the T2
sampling location because the soil was too well protected by the thorny, dense crown. Tree-tree distances on test sites with a high tree density could be very small, so that IT3 and IT4 sampling locations were the same on these test sites. Mean tree-tree distances on the test sites varied from 1.8 m to 17.5 m, with trees sometimes growing in tree groups and not being equally distributed on the test sites, so that some distances between trees could be much larger with a maximum of 50 m on one test site.

For more information about the potentially different conditions of tree and intertree area near the tree drip line, the unsaturated hydraulic conductivity of the soils was chosen. Dual measurements with a tension-disc infiltrometer were taken in one direction parallel to the contour lines at the tree drip line at the T2 and IT3 locations for 19 out of 30 test sites, with 74 measurements overall in October/November 2019. For each test site we measured T2 and IT3 simultaneously on the same day to be able to compare the two sampling locations (see section 2.4 for more information).

**2.3 Soil analyses**

The pH was measured using distilled water (DIN ISO 10390:1997-05, 1997). The electric conductivity was determined by the method described in DIN ISO 11265:1997-06 (1997). The measurement of the pH-value and the electrical conductivity was performed using the *WTW Multi 3410 Set Tetra Con*.

To measure the percolation stability, the method described by Auerswald (1995) and Becher (2001) was used. 10 g of air-dried
aggregates, sieved for the size 1-2 mm, were prepared in plexiglass tubes enclosed by a thin layer of sand on each end. To acquire a homogenous and tight packing of the material the tube was tapped 20 times onto a hard surface. The tube was connected to a bottle of water placed on top of a scale which recorded the amount of water flowing through the tubes with a constant water pressure of 20 hPa, which was maintained by a Mariotte's bottle. The experiment lasted 10 minutes, higher amounts of water passing through the tubes implied a higher stability of the aggregates, as water still found its way through
macropores, while lower values showed a low stability as aggregates broke apart and water could only flow through fine pores. The outcomes were corrected for total sand according to the equation described by Mbagwu and Auerswald (1999).

The total soil carbon content was analysed using thermal oxidation and infrared spectral detection using the carbon analyser LECO RC-412. The content of organic and mineral carbon was determined by a combustion in two steps with temperatures of 550 °C and 1000 °C. The total nitrogen content was analysed with the TruSpec Macro by Leco in accordance with DIN EN
16168:2012-11 (2012) with a temperature of 950 °C. Both soil organic carbon (SOC) and nitrogen (TN) content are given in %.

**2.4 Tension-disc infiltrometer experiments**

The unsaturated hydraulic conductivity was measured by tension-disc infiltrometers by Decagon Devices, METER Group Inc. (METER Group Inc., Munich, Germany). This infiltrometer is divided into two chambers. The device uses the principle of a
Mariotte's bottle with the two chambers being connected by two small tubes and the lower chamber being contained by a sintered-steel disc. One of the tubes can be used to adjust the suction, thus being able to eliminate the flow through macropores (Dohnal et al., 2010). This device's steel disc has a diameter of 4.5 cm with a water capacity of 135 ml in the lower chamber. Tap water was used for the experiment. Since the soils in the study areas are very heterogenous with many embedded rocks in the soil surface, the hydraulic contact between the soil and the tension-disc infiltrometer was ensured with a thin layer of sand
between the disc and the soil (Hopmans et al., 2002; Perroux and White, 1988; Reynolds and Zebchuk, 1996). Each measurement consisted of four runs to ensure measuring hydraulic conductivity with different suction rates, namely 4, 2, 1 and

0.5 cm. Each run lasted 15 minutes and infiltration was measured every minute in the beginning to every three minutes in the end of the experiment. We used the method of Zhang (1997) to determine hydraulic conductivity by using Eq. (1):

$$K = \frac{C_1}{A},$$
(1)

where K equals the hydraulic conductivity, $C_1$ equals the slope of cumulative infiltration over time and A equals a value putting the van Genuchten parameters of the measured soil in relation to the chosen suction rate and the radius of the infiltrometer disc (van Genuchten, 1980). Carsel and Parrish (1988) provide van Genuchten parameters for 12 different texture classes. At the time of the measurements in October/November 2019 soils were very dry in all three study areas (soil water content at the measurement points: 0.1 – 0.6 %). Soil texture classes were the same for T2 and IT3 sampling locations on 16 out of 19 test

sites.

## 2.5 Statistical analyses

Potential differences between T1, T2, IT3 and IT4 were tested using a Kruskal-Wallis-Test. In case of significant differences, subsequent post-hoc tests (Dunn-Bonferroni-Tests) were carried out to find which groups differed significantly from each other (p < 0.05). A Wilcoxon-Test was used to test for potential differences between the tension-disc infiltrometer

measurements for T2 and IT3 sampling locations. These tests were carried out using the software IBM SPSS Statistics 25 (IBM, Armonk, USA). Since significance tests only show if there are differences in the data but do not give information about the size of the difference, the effect size was calculated. Because the p-value is dependent on the size of the sample as well as the size of the effect, it is possible to receive a significant result with a large enough sample but a small effect (Coe, 2002). Thus, the effect size is used to quantify the difference between the data if a significant result is found. In this study Pearson's

r was used as the effect size, where the values 0.1, 0.3 and 0.5 show a small effect, a medium effect and a large effect respectively (Cohen, 1988; Cohen, 1992).

## 3 Results

### 3.1 Differences between the sampling locations T1 - IT4

The differences between the tree area (T1) and the intertree area (IT4) have been previously discussed by Kirchhoff et al.

(2019a). However, in the previous study the samples were only taken in one direction and did not take the tree drip line into account (T2 and IT3). Table 1 displays the average values of T1, T2, IT3, IT4 for the studied soil parameters regardless of the direction from the tree.

**Table 1: Mean and standard deviation values for analysed parameters for T1 (near tree trunk), T2 (near tree drip line, within tree**
**area), IT3 (near tree drip line, outside tree area) and IT4 (intertree area between two trees) samples. EC: electrical conductivity; PS: percolation stability; TN: total nitrogen content; SOC: content of soil organic carbon; C/N: C/N ratio.**

| Parameter/ Sample | Fine material < 2 mm (%) | Coarse material > 2 mm (%) | Soil water content (%) | pH | EC (µS) | PS (ml 10 min$^{-1}$) | TN (%) | SOC (%) | C/N |
|---|---|---|---|---|---|---|---|---|---|
| **T1** | 79.0 ± 14.3 | 21.0 ± 14.3 | 1.2 ± 0.8 | 7.9 ± 0.2 | 306.8 ± 77.8 | 190.6 ± 107.4 | 0.3 ± 0.2 | 4.4 ± 2.5 | 12.4 ± 3.6 |
| **T2** | 73.8 ± 14.6 | 26.2 ± 14.6 | 0.8 ± 0.7 | 7.8 ± 0.2 | 268.0 ± 38.6 | 148.2 ± 123.3 | 0.2 ± 0.1 | 2.0 ± 1.0 | 9.3 ± 2.4 |
| **IT3** | 71.8 ± 14.8 | 28.2 ± 14.8 | 0.7 ± 0.7 | 7.8 ± 0.2 | 256.2 ± 54.1 | 90.5 ± 84.2 | 0.2 ± 0.1 | 1.3 ± 0.7 | 8.1 ± 2.1 |
| **IT4** | 72.1 ± 13.6 | 27.9 ± 13.6 | 0.5 ± 0.8 | 7.8 ± 0.2 | 235.1 ± 29.6 | 60.5 ± 72.1 | 0.1 ± 0.0 | 1.0 ± 0.6 | 7.9 ± 3.8 |

The averages show that there is a continuous decline of values from T1 along T2 and IT3 to IT4 for the parameters soil water content, EC, PS, TN, SOC and C/N. The content of fine material is highest in the tree area, yet has decreased from the tree trunk to the tree drip line. Outside the canopy the content of fine material shows the lowest values while the content of coarse material is likewise increasing. A Kruskal-Wallis-Test confirmed that there are significant differences ($p < 0.05$) depending on the sampling position for the parameters soil water content, pH, EC, PS, TN, SOC and C/N. These parameters were subsequently analysed for significant differences using Dunn-Bonferroni-Tests and for their effect size (Tab. 2). Table 2 shows that most differences between the sample points are significant with the exception of pH-values (n/a), while fine and coarse material do not show significant differences at all. The differences between T1 and T2 are not significant for soil water content, EC and PS, thus indicating an influence of the canopy cover on these parameters. Soil water content is also not significantly different near the tree drip line (T2, IT3) suggesting a possible influence of the tree on the intertree area. Large effect sizes are visible for the parameters PS, TN, SOC and C/N and show the large difference of values between the T1 and IT4 sample locations. This is visible for soil water content and EC as well but with only medium effect sizes. TN, SOC and C/N also show significant differences with a large effect between T1 and IT3, a medium effect between T2 and IT3, while the difference between T2 and IT3 shows only small to medium effects, indicating that the closer the sample is located to the tree trunk, the higher the value will be. Boxplots from T1-IT4 sampling locations are shown exemplarily for SOC (Fig. 4) with highest values around the trunk and PS (Fig. 5) with a high amplitude of values in the T2 sampling location.

**Table 2: Effect sizes (Pearson's r) for significant differences (p < 0.05) between sampling locations for the analysed parameters. Values show the size of the effect, if significantly different. n/a = not applicable, no significant difference; small effect ≥ 0.1; medium effect ≥ 0.3 (in italics and underlined); large effect ≥ 0.5 (bold). EC: electrical conductivity; PS: percolation stability; TN: nitrogen content; SOC: content of soil organic carbon; C/N: C/N ratio.**

| Parameter/ Comparison | Soil water content | pH | EC | PS | TN | SOC | C/N |
|---|---|---|---|---|---|---|---|
| **T1 – T2** | n/a | *0.37* | n/a | n/a | *0.3* | *0.34* | *0.35* |
| **T1 – IT3** | 0.28 | 0.29 | *0.31* | *0.39* | **0.53** | **0.6** | **0.56** |
| **T1 – IT4** | *0.43* | n/a | *0.49* | **0.55** | **0.72** | **0.77** | **0.68** |
| **T2 – IT3** | n/a | n/a | 0.19 | 0.24 | 0.29 | *0.33* | 0.26 |
| **T2 – IT4** | 0.29 | 0.21 | *0.42* | *0.45* | **0.53** | **0.55** | *0.42* |
| **IT3 – IT4** | 0.18 | n/a | 0.23 | 0.21 | 0.24 | 0.22 | n/a |

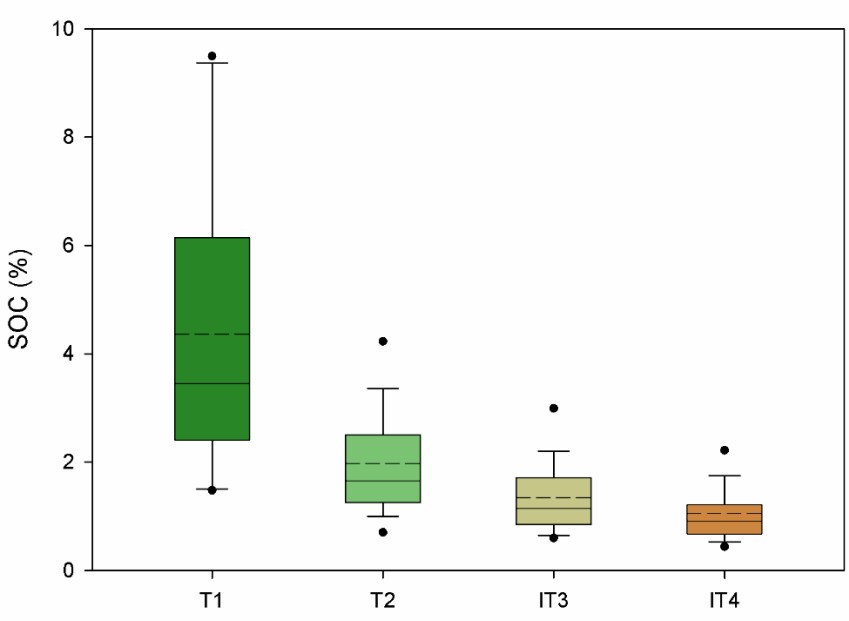


**Figure 4: Boxplots showing the content of soil organic carbon (SOC) for T1 (near tree trunk), T2 (near tree drip line, within tree area), IT3 (near tree drip line, outside tree area) and IT4 (intertree area between two trees) sampling locations. Median: solid line, mean: dash line, dots: 5th and 95th percentile.**

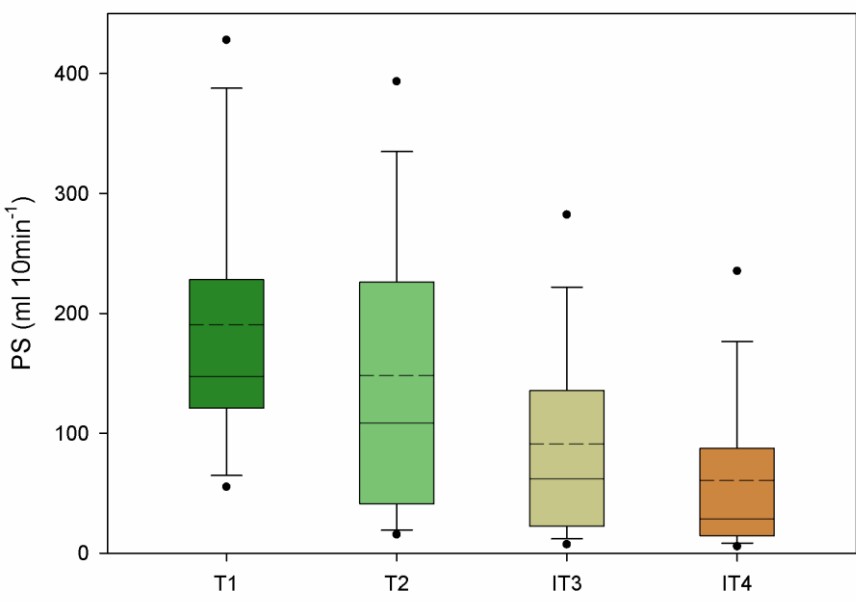


**Figure 5: Boxplots showing the results of percolation stability (PS) analysis for T1 (near tree trunk), T2 (near tree drip line, within tree area), IT3 (near tree drip line, outside tree area) and IT4 (intertree area between two trees) sampling locations. Median: solid line, mean: dash line, dots: 5th and 95th percentile.**

A possible correlating factor with the T1-IT4 results is the distance from the tree trunk, yet it did not yield any significant correlating results. A normalisation of this distance with the crown's radius in the measured direction showed an $R^2 = 0.27$ for an exponential trend line for the parameter SOC (no correlation for the other parameters), yet showed no significance in the results.

### 3.2 Tension-disc infiltrometer experiments

The unsaturated hydraulic conductivities ($K_h$) near the tree drip line (T2 and IT3) were measured on 19 test sites. Figure 6 compares the measurement locations T2 and IT3 with the different suction rates used (4, 2, 1 and 0.5 cm). The $K_h$-values increase from higher to lower suctions, since the water is able to infiltrate into coarser pores and more water can infiltrate into the soil. T2 shows average values of 17.6, 26.5, 33.2 and 41.1 mm h$^{-1}$ for the suction rates 4, 2, 1 and 0.5 cm respectively. The mean values of IT3 for the same suction rates are 12.6, 19.6, 24.6 and 29.6 mm h$^{-1}$. The measurement location T2 differs

significantly from IT3 ($p < 0.01$ for suction rate 4, 2, 0.5 cm and $p < 0.05$ for suction rate 1 cm). The effect sizes are r=0.51 for suction rate 4 cm, showing a high effect, while suction rates 2, 1 and 0.5 cm show medium effect sizes with r=0.49, 0.42 and 0.47, respectively.

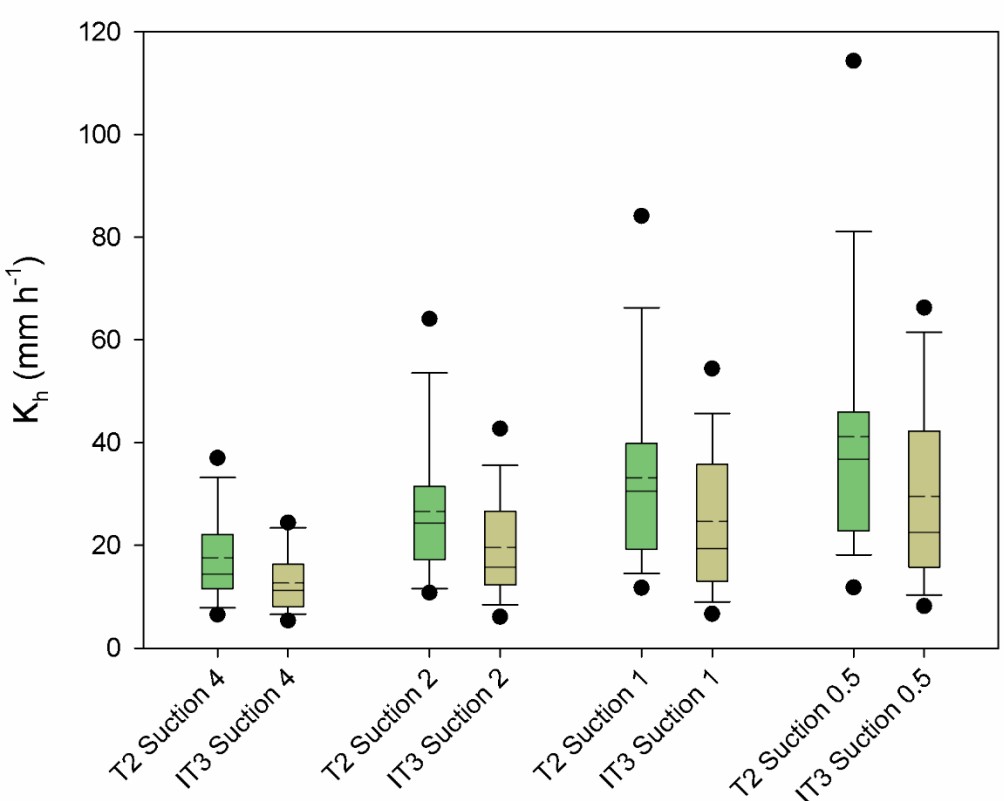

**Figure 6: Boxplots showing the unsaturated hydraulic conductivities ($K_h$) values for suctions 4, 2, 1 and 0.5 for T2 (near tree drip line, within tree area) and IT3 (near tree drip line, outside tree area) sampling locations. Median: solid line, mean: dash line, dots: 5th and 95th percentile.**

### 3.3 Directional patterns

Averaging the soil-parameter values over all directions for T2, IT3 and IT4 has shown a decrease of values from tree trunk to intertree area. However, if there were an influence of the tree in one specific direction, the means in Tab. 1 would not show it. All directions of sampling (N, NE, E, SE, S, SW, W and NW) need to be looked at separately.

We drew spider diagrams to show if the values were distributed equally along all directions (Fig. 7). The parameters SOC and TN show higher T2 mean values in the E to S directions. In the IT3 mean values this trend is visible but not as pronounced while the IT4 values only show a slight tendency in the S direction. The parameter soil water content shows the highest values in N and SE directions for the T2 and IT3 sampling locations, EC and C/N show very similar values along all directions but a slight tendency in E to S directions for the T2-samples. PS shows the highest means for T2 in the E and NW directions, for the

IT3 samples NW and SE directions are the most pronounced. The average pH-values appear highest in the NE, SE, SW and NW directions for the IT4 samples.

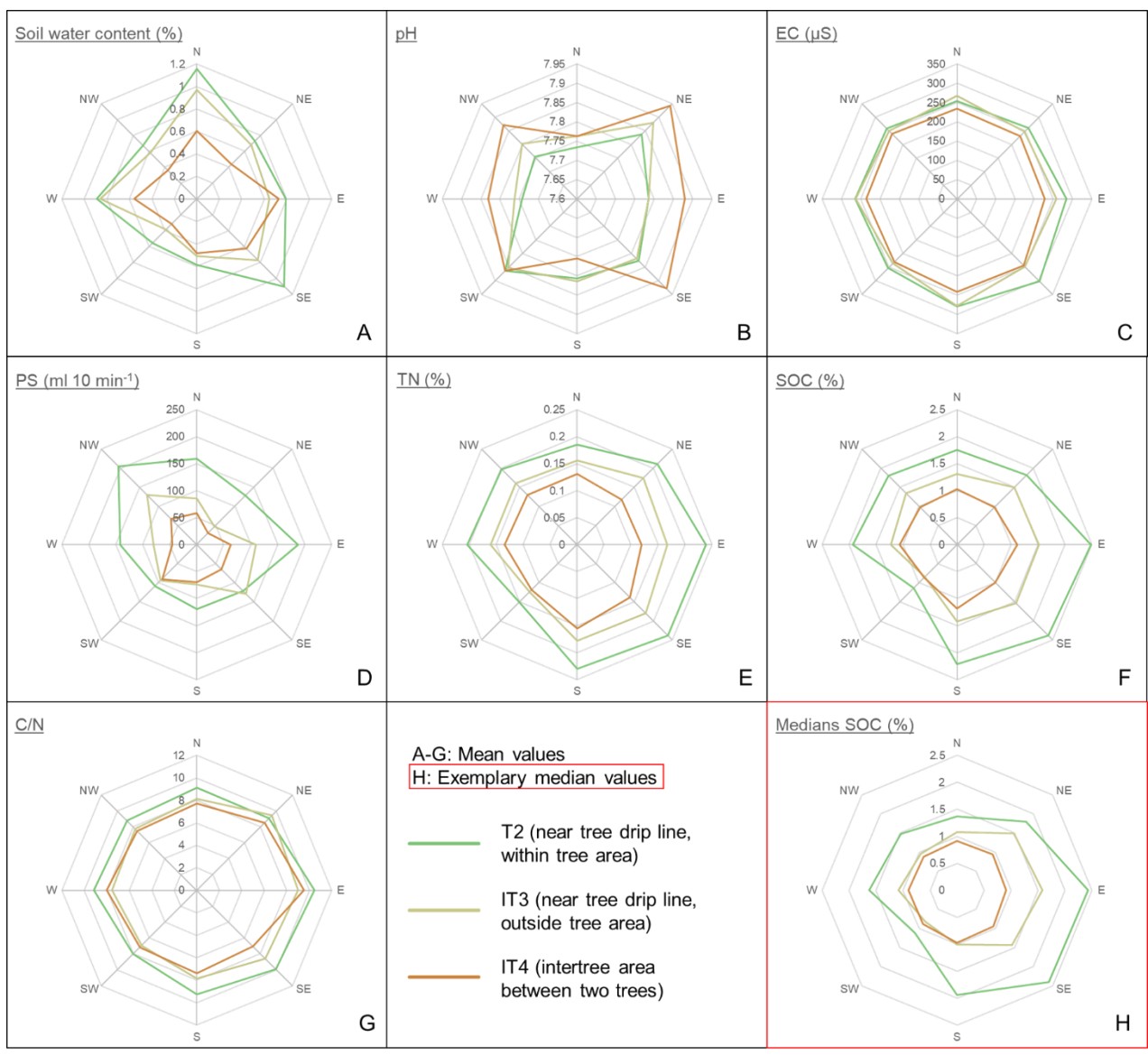

Figure 7: Spider charts showing the means for each direction and parameter and sampling location. A: Mean soil water content, B: mean pH, C: mean electrical conductivity, D: mean percolation stability, E: mean total nitrogen content, F: mean content of soil organic carbon, G: mean C/N ratio, H: median content of soil organic carbon.

Since not all test sites were situated on equally exposed slopes, we also normalised the cardinal directions of the data to show the directions downslope, upslope and along the contour lines to the left and right (looking upslope) from the tree. When the

data is thus reorganized, it becomes apparent that SOC (exemplary) is distributed further downslope or to the right side of the tree (Fig. 8). In most cases eastern and southern directions are at the right side and downslope, respectively. Upslope corresponds to the direction N in 19 out of 30 cases, right corresponds to the direction E in 16 out of 30 cases, downslope to S in 15 out of 30 cases and left to W in 16 out of 30 cases.

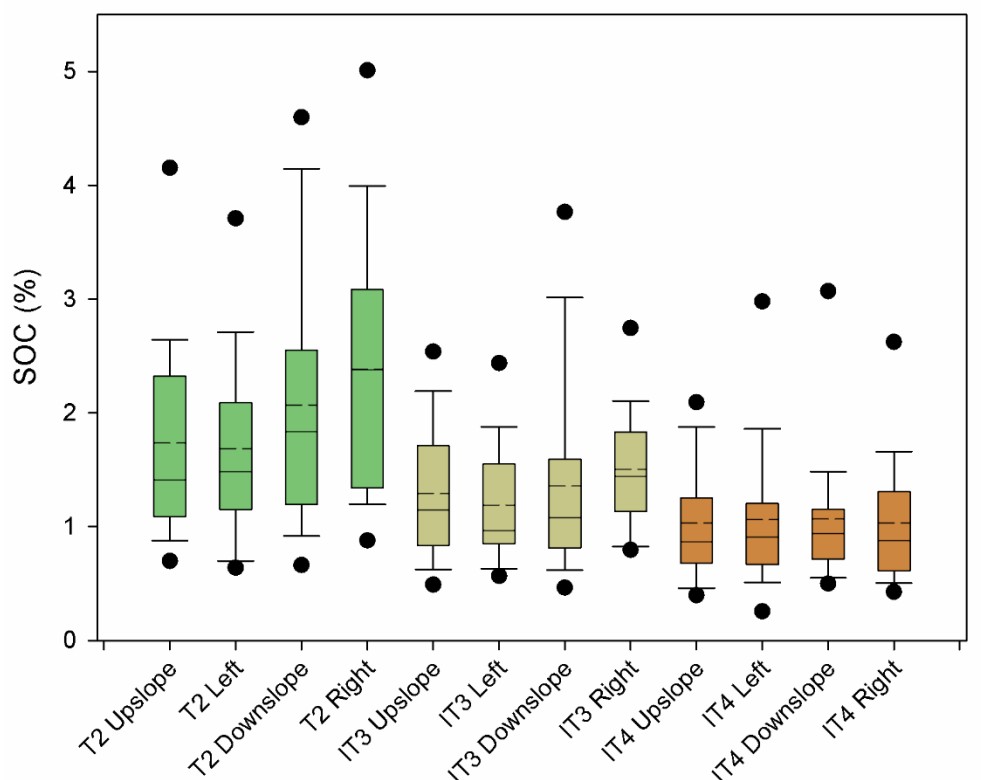

**Figure 8: Boxplots showing the content of soil organic carbon (SOC) values for slope positions upslope, left of the tree looking upslope, downslope, right of the tree looking upslope for sampling locations T2 (near tree drip line, within tree area), IT3 (near tree drip line, outside tree area) and IT4 (intertree area between two trees). Median: solid line, mean: dash line, dots: 5th and 95th percentile.**

Table 3 shows the averages plus standard deviations for all parameters for each slope direction. The parameters EC, PS, SOC and C/N show the highest averages for T2 right of the tree. PS shows the highest means downslope for IT3 and IT4, while soil water content shows the highest means upslope and left for T2 and upslope and right for IT4. Test sites on steeper slopes do not show higher translocation of material downslope as test sites with lower slope values in all cases.

**Table 3: Average values ± standard deviations for all parameters for each sample location in slope direction. T2: near tree drip line, within tree area; IT3: near tree drip line, outside tree area; IT4: intertree area between two trees; EC: electrical conductivity; PS:**

295 percolation stability; TN: total nitrogen content; SOC: content of soil organic carbon; C/N: C/N ratio. The singular highest values for each parameter for T2/IT3/IT4 are highlighted bold.

| Parameter/ sample location | | Soil water content (%) | pH | EC (µs) | PS (ml 10min⁻¹) | N (%) | SOC (%) | C/N |
|---|---|---|---|---|---|---|---|---|
| T2 | upslope | 0.9 ± 0.7 | 7.7 ± 0.2 | 254.7 ± 31.5 | 151.8 ± 138.1 | 0.2 ± 0.1 | 1.7 ± 0.9 | 9.1 ± 2.1 |
| | left | 0.9 ± 0.7 | **7.8 ± 0.1** | 261.9 ± 38.8 | 142.2 ± 124.6 | 0.2 ± 0.1 | 1.7 ± 0.8 | 8.7 ± 1.8 |
| | downslope | 0.8 ± 0.8 | 7.8 ± 0.2 | 271.6 ± 36.6 | 137.6 ± 104.6 | 0.2 ± 0.1 | 2.1 ± 1.1 | 9.4 ± 2.7 |
| | right | 0.7 ± 0.6 | 7.8 ± 0.2 | **283.2 ± 41.5** | **167.1 ± 122.3** | 0.2 ± 0.1 | **2.4 ± 1.1** | **9.9 ± 2.6** |
| IT3 | upslope | 0.7 ± 0.7 | 7.8 ± 0.2 | 249.2 ± 36.7 | 84.0 ± 74.2 | 0.2 ± 0.1 | 1.3 ± 0.6 | 8.5 ± 2.7 |
| | left | 0.7 ± 0.7 | 7.8 ± 0.2 | **260.3 ± 66.4** | 87.2 ± 79.7 | 0.2 ± 0.0 | 1.2 ± 0.5 | 7.6 ± 1.4 |
| | downslope | 0.7 ± 0.7 | 7.8 ± 0.2 | 250.5 ± 26.1 | **102.6 ± 100.6** | 0.2 ± 0.1 | 1.4 ± 0.9 | 7.9 ± 2.1 |
| | right | 0.6 ± 0.6 | 7.8 ± 0.2 | 252.8 ± 28.4 | 91.2 ± 78.3 | 0.2 ± 0.0 | **1.5 ± 0.5** | **8.5 ± 1.9** |
| IT4 | upslope | **0.6 ± 0.7** | 7.8 ± 0.2 | 234.9 ± 29.1 | 54.8 ± 58.4 | 0.1 ± 0.0 | 1.0 ± 0.5 | 7.8 ± 2.2 |
| | left | 0.5 ± 0.6 | 7.9 ± 0.2 | 233.4 ± 31.6 | 65.6 ± 78.4 | 0.1 ± 0.0 | 1.1 ± 0.6 | **8.4 ± 5.6** |
| | downslope | 0.5 ± 0.7 | 7.8 ± 0.2 | **236.7 ± 31.4** | **80.4 ± 87.1** | 0.1 ± 0.1 | 1.1 ± 0.6 | 7.4 ± 1.2 |
| | right | 0.6 ± 1.0 | 7.9 ± 0.2 | 235.1 ± 27.4 | 46.2 ± 57.0 | 0.1 ± 0.0 | 1.0 ± 0.6 | 8.2 ± 4.4 |

However, a Kruskal-Wallis-Test on the geographic directions as well as on the slope directions did not yield significant differences. Trends in specific directions are visible, but not every tree shows the same directional bias for each parameter.

300 **4 Discussion**

Despite being canopy-covered, the distance from the trunk to the tree drip line shows a decrease of values for most analysed soil parameters. A significant difference was found between IT3 and IT4 values for most parameters, yet showed only a small effect. The distance from the tree trunk as well as the normalised distance by the crown's radius was considered to be a correlating factor, yet did not show any significant correlation to the studied parameters. This might be because there is not an
305 even distribution of litter cover along all directions and due to the difference in the test sites.

The decrease of parameter values from under the tree crown via the tree drip line to the intertree area was discussed before by Zinke (1962) for much different climatic conditions and forest densities. Analyses on the effects of savanna trees (Belsky et al., 1993) and acacia trees (De Boever et al., 2015) found a rapid decrease of organic matter from the tree trunk in most cases,

which corresponds well with our findings for argan trees. A decrease of nitrogen content with lateral distance from the plant was also described by García-Moya and McKell (1970). The better soil properties under trees or shrubs in comparison to the corresponding intertree or intershrub area have been discussed in fertile island research before (Boettcher and Kalisz, 1990; Pérez, 2019; Qu et al., 2018). Garner and Steinberger (1989) argued that micro-, meso- and macro-fauna are drawn to the tree area since it contains the highest soil water content, the lowest day-time temperatures and the highest abundance of food sources. This concentration is the cause for the fertile island structure in arid environments. The area around the tree drip line under the canopy is exposed to the sun at least part of the day while the amount of litter is not as high as around the trunk. This could be a reason for the much lower values at the T2 sampling locations.

The average values of T2 and IT3 differ significantly for all analysed soil parameters as well as for unsaturated hydraulic conductivities but with different effect sizes. However, the difference between T2 and IT3 is much lower than the difference between T1 and T2 for most parameters. As seen in Fig. 7, the IT3 values are more similar to the IT4 values than to the T2 values. The results lead to the conclusion that the influence of the tree on the intertree area is limited. De Boever et al. (2014) found that organic material correlated well with soil porosity and bulk density thus possibly explaining the medium to large effect for the unsaturated hydraulic conductivities between T2 and IT3 sampling locations. Another possible explanation is the higher number of broken-up aggregates due to splash erosion and a subsequent sealing of the pores outside of the tree area (le Bissonais, 1996). In a previous study, we found higher average suspended sediment concentrations (4.42 g $L^{-1}$ compared to 2.18 g $L^{-1}$ under argan trees) as well as lower average infiltration rates in the intertree areas (229.56 mm $h^{-1}$ compared to 452.57 mm $h^{-1}$ under argan trees) (Kirchhoff et al., 2019a). However, the unsaturated hydraulic conductivities of the T1 and IT4 in our earlier study should not be compared with the T2 and IT3 values in this study, since they were sampled in two different field phases under different seasonal conditions. Since all suction rates displayed significant differences with medium to large effects between T2 and IT3 sampling locations, we can assume that it is harder for water to infiltrate into the soil outside the tree drip line. This leads to the conclusion that erodibility is higher outside the tree drip line as well (Peter and Ries, 2013) which is confirmed by the percolation stability values that are also closely linked to erodibility (Auerswald, 1995; Mbagwu and Auerswald, 1999) and mostly show values < 250 ml 10 $min^{-1}$ that would lead to higher interrill erosion, even in the tree area (Mbagwu and Auerswald, 1999).

Although no significant differences between the sampling directions were found, trends are visible and can be attributed to the processes acting on the tree and the surrounding intertree areas. The parameter SOC shows the most pronounced effects in eastern directions (mostly E, SE) which corresponds to the main wind direction. Marzen et al. (2020) found relatively high wind erosion rates under trees using an experimental wind tunnel on one of the here-analysed study sites with the eroded material being mostly tree litter. Sirjani et al. (2019) show a negative correlation between SOC-content and wind erosion, since higher organic matter often leads to more particle aggregation and more stable aggregates (Tatarko, 2001). However, Chepil (1954) argues that these aggregates are not big enough to resist the erosive winds common in drylands. As is shown in Tab. 3, litter is mostly translocated to the right side of the tree (view upslope, T2 and IT3) due to its lighter weight while more stable aggregates are found on the right side of the tree (T2) but downslope for IT3 and IT4. This suggests that litter is mostly moved

by wind from the tree to the intertree area but that larger soil particles and aggregates are further translocated by slope wash. Although the percolation stability is measured on 1-2 mm macroaggregates (Auerswald, 1995) it is reasonable to assume that smaller aggregates are similarly stable and wind can move them by surface creep to the east only small distances (Chepil, 1945; Lyles, 1988; Yang et al., 2020). Dunkerley (2000) remarked that litter was likely to be washed out and dispersed leading to low amounts of litter in the intertree areas, which is visible in Tab. 3. Some test sites show higher IT4 values in the west (or right side of the tree, e.g., C/N in Tab. 3), suggesting a translocation of litter further from the tree drip line into the intertree areas. Since tree-to-tree distances vary, the size of the intertree areas vary as well, with close-spaced IT4s showing higher values than wide-spaced intertree areas, where translocation from the tree area is less likely (Li et al., 2008; Zhang and Wang, 2017). Trees in close distance from each other could lead to a reduction in wind speed in the intertree area, yet for most trees the eroded material (mostly lightweight litter) is deposited in the quiet zone behind the tree (Leenders et al., 2007).

The type of tree (architecture, size, genetic variety) could be a possible explanation for the missing significance of the directions. Since argan trees differ in their architecture due to degradation by overbrowsing and woodcutting (Culmsee, 2005; le Polain de Waroux and Lambin, 2012) they also differ in their potential to shield the soil under the trees from wind or water erosion. Very degraded shrub-like trees should protect the canopy-covered area much better than tall trees whose crown is not in contact with the surface. The litter production can also be a determining factor for the differences between the analysed trees. As Zahidi et al. (2013a) point out, some trees shed their leaves during long periods of drought, possibly leading to a higher production of litter that can be blown or washed off. As different morphotypes and genotypes exist (Majourhat et al., 2008), differences in soil parameters might not originate only from the test sites themselves but also the differences of the sampled trees.

Although most of the measured soil parameter values decrease from T1 to T2, the T2 values might still be high enough to support young seedlings. Defaa et al. (2015) found higher chances of seedling survival when planted near tree shelters, possibly because of a better microclimate, which matches the soil water content values around the tree measured in this study. Higher soil qualities were also found in the intertree areas near fertile islands (Qu et al., 2018), while degradation could be halted by short-rotation forestry of Eucalyptus in NW Morocco (Boulmane et al., 2017). However, feedback processes could lead to a transition to a fully arid ecosystem (Schlesinger et al., 1990), especially under the current land use pressure, making it increasingly difficult for reforestation. This is visible for some parameters in the relatively high differences between T1 and T2 values. Climate change will also make parts of the current habitat unsuitable for *Argania spinosa* (Moukrim et al., 2019), with future droughts making it necessary for human intervention to reduce damage to seedlings (Chakhchar et al., 2017; Zahidi et al., 2013b) by measures to prevent overgrazing.

## 5 Conclusion

In this study, we were able to show that spatial patterns of tree influence on soil properties exist around argan trees. They are most pronounced due to, e.g., litter translocation to the east and downslope of the tree and due to the effect of midday shade

to the north of the tree on gravimetric soil water content. The decrease of the studied soil parameter values with increasing distance from the tree trunk suggests that a fertile island structure is concentrated solely around the trunk while the rest of the tree area is still protected by the canopy, as visible in the infiltration properties, but is more similar to the intertree area. Reforestation measures should aim to plant young sprouts close to the trees, ideally in northern or eastern directions to take advantage of the shade (higher soil water content) and the higher content of soil organic carbon and nitrogen. Since not all

argan trees are similar in size, tree architecture and genetic variety, more research is needed on how these factors influence the soil under argan trees.

## Data Availability

The data of the soil analyses and tension-disc infiltrometer experiments from this study are available upon request.


## Author Contributions

M. K. and T. R. performed the investigation and wrote the original draft. M. K. performed data curation. I. M. reviewed the paper. I. M., M. S., A. A. H. and J. B. R. provided resources and supervised this research.

**Competing interests**

The authors declare no conflicts of interest.

## Acknowledgments

This research was funded by Deutsche Forschungsgemeinschaft (DFG), grant numbers RI 835/24-1 and MA 2549/6-1. We
hereby acknowledge the support of the laboratories of Goethe University Frankfurt am Main, Trier University as well as of the Université Ibn Zohr Agadir. We also would like to thank Tobias Buchwald, Lars Engelmann, Laura Kögler and Lutz Leroy Zimmermann for their considerable help during data acquisition in the field and laboratory as well as Oliver Gronz for his advice on data analysis and graphical implementation.

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
