# Peer review of "Spatial distribution of argan-tree influence on soil properties in South Morocco"

_SOIL, 2021_

## Author Response (AR1)

Dear reviewers,

thank you for the comprehensive and constructive reviews. We have made changes to the manuscript according to your comments. You can find answers to the comments further down point by point, our responses are highlighted in italic. Thank you very much for your time.

Referee 1

GENERAL COMMENTS

The manuscript ID soil-2021-32 shows interesting results from an empirical study carried out in southern Morocco that compares values of soil quality between different spatial areas previously dominated by argan trees. From a formal point of view, the article is quite well-written (although not well-structured in the M&M section), the methodology is consistent, conclusions are supported by its results and the literature background is relatively wide. Regarding novelty, the most interesting of this research is the provision of data from a geographical are not deeply studied yet. In addition, authors have made a remarkable effort in fieldwork and analysing data. So, I am going to suggest authors make some changes aimed at improving the rigor and quality of the manuscript previous to suggest my acceptance. Please see below my specific comments. I hope they can be helpful. Of course, my good feeling with this manuscript maybe is not shared by the opinions expressed both by other reviewer(s) and editor(s). In other words, it is just my opinion on this research/article.

SPECIFIC COMMENTS

Title

Soil quality is much more than the simple use of some parameters. I am not 100% sure if soil quality is the right concept to be included in the title. When a regular reader reads soil quality he/she probably thinks about the use of indicators, indexes, assessment systems, etc. Anyway, it is a personal decision that must be made by the authors. I have only expressed my opinion about it.

*Response: Thank you for this comment. We agree that the term "soil quality" is misleading and have changed the title accordingly to: „Spatial distribution of argan-tree influence on soil properties in South Morocco". We have also changed this issue in the text.*

Abstract

This section should be composed by a single and compact paragraph. Perhaps the guidelines of the journal ask for using a paragraph for each idea (motivation, methodology, outstanding results, and main conclusion). In that case, please follow the guidelines but I prefer the traditional way.

*Response: Thank you for pointing that out to us. We changed the abstract into one single paragraph.*

Lines 9-10 (motivation): In my opinion, this sentence is quite logical and obvious, except in overgrazed areas (litter remotion) or in particular cases. Bare soil surfaces usually tend to

reduce soil quality in comparison with nearby tree-influenced areas. Please rewrite or remove it.

*Response: We deleted the subclause, but are keeping the main clause, since it is important to point out, that there is limited regrowth in argan woodlands and thus the intertree areas increase. It now says: "Bare areas between the isolated trees increase due to limited regrowth, however, it is unknown if the trees influence the soil of the intertree areas." (lines 9-10 in the revised manuscript).*

Lines 13-14 (methodology): Please provide information about the maximum distance from the trunk where you collected soil samples. For instance, I know works in which authors have detected influence of trees on soil nutrient contents at a distance of 8 m from the trunk (open area).

*Response: We added the maximum distance (50 m) and some more information in the experimental design section. Since our test sites vary regarding tree density, there are some with large distances between trees and some with very small distances. Mean distances between the trees varied from 1.8 m to 17.5 m per test site, however trees sometimes grew in groups and were not evenly distributed on the test sites which is why some distances between trees could be much larger.*

Line 15: Soil moisture? At field capacity? When have you measured it? Please mention it.

*Response: We have changed soil moisture to gravimetric soil water content throughout the text. We mention the sampling period in line 147 (February/March 2019), and measured the gravimetric soil water content by weighing and drying the soil.*

Line 18: I suggest using SOC instead of C-org and "and" instead of "&" (throughout the text). I think it is more rigorous.

*Response: Thank you for this comment. We changed it into both „SOC" and „and" throughout the text.*

Lines 18-23: I miss some brief information about the sampling plots, i.e. to know which kind of areas have been compared since I only know some properties have been assessed in 385 points following transects around the argan-trees.

*Response: We added a sentence: "This was the case on sites under different land usages (silvopastoral, agricultural), slope gradients or tree densities." (lines 23-24 in the revised manuscript).*

Line 25: Along with reforestation I think measures to prevent overgrazing, illegal firewood extraction, etc. should be also mentioned as friendly solutions.

*Response: Thank you. We added a phrase at the end: "...ensure seedling survival, along with measures to prevent overgrazing." (lines 26-27 in the revised manuscript).*

Introduction

The length of this section could be reduced.

*Response: We reduced the section by some sentences.*

Broadly speaking, this section is well-written and organized and it is easy to read and understand. It follows a logical order touching every point interesting for this kind of studies. Regarding literature (and its citations), in my opinion, authors have cited too many references to support each idea and I have missed some important references of works made in similar areas. Nevertheless, it is something personal and authors must feel free to use those references they consider pertinent. Anyway, citations must be sorted in alphabetical or in temporal order. None of them has been respected.

*Response: Thank you for this comment. We sorted all citations alphabetically and deleted some references, where many were cited, while we added some references to introduction and discussion: García-Moya and McKell, 1970, Ludwig et al., 2005 and Yang et al., 2020 (lines 34, 311, 347 in the revised manuscript).*

*García-Moya, E., and McKell, C. M.: Contribution of Shrubs to the Nitrogen Economy of a Desert-Wash Plant Community, Ecology, 51(1), 81-88, 1970.*

*Ludwig, J. A., Wilcox, B. P., Breshears, D. D., Tongway, D. J., and Imeson, A. C.: Vegetation patches and runoff-erosion as interacting ecohydrological processes in semiarid landscapes, Ecology, 86(2), 288-297, doi: 10.1890/03-0569, 2005.*

*Yang, C., Geng, Y., Fu, X. Z., Coulter, J. A., and Chai, Q.: The Effects of Wind Erosion Depending on Cropping System and Tillage Method in a Semi-Arid Region, Agronomy, 10, 732, doi:10.3390/agronomy10050732, 2020.*

Lines 42-45: This paragraph is too short. Please think also about the reading of the manuscript.

*Response: We have merged both paragraphs about argan woodlands into one.*

Line 43: agro? Please provide more details. I guess some cereals are temporally cropped in this kind of land systems.

*Response: Cultivation of cereals between trees is common in argan woodlands. We have added: "… rainfed cultivation of cereals between the trees (highly speculative due to high variation in precipitation) and collection…" (lines 46-47 in the revised manuscript).*

Figure 1 is wonderful but I think figures should not be presented in an Introduction section. Perhaps it could be moved to the M&M section and sampling points could be also drawn.

*Response: We moved Figure 1 (now Figure 2) to the section Experimental design, with a note in the introduction to see 2.2. Experimental design. We prefer having two figures, one for the hypothetical influences and one for the sampling scheme, to not overload the figure.*

Material and methods

I suggest call this section as Material and methods instead of Material & Methods.

*Response: Thank you. We have changed it.*

This section should be fully restructured. Please see below some of my comments.

*Response: Thank you for the comments and solutions for restructuring the Material and methods section. We restructured it the following way. 2 Material and methods, 2.1 Study areas, 2.2 Experimental design (with figures for potential influences and soil sampling), 2.3 Soil analyses, 2.4 Tension-disc infiltrometer experiments, 2.5 Statistical analyses.*

Experimental design must be perfectly explained. I have had to read several times this section to understand this. A regular reader will only read once and he/she must understand it after the first reading. Please concentrate your effort to improve this weakness.

*Response: Thank you for pointing this out. We changed the explanation to: „To measure the potential influence of the tree on the intertree area shown in Fig. 2, we took 13 samples around each tree. One was taken next to the trunk, while we took three soil samples each in four directions around the tree, namely upslope, downslope and in both directions parallel to the contour lines. The three samples in each direction were taken with increasing distance from the tree, one near the tree drip line under the canopy, the next near the tree drip line just outside the crown's cover, and the third in the intertree area at the midpoint between the tree and its next neighbouring tree in that direction (Fig. 3). The two samples at the tree drip line were generally about one metre apart, depending on the crown's shape and the surface conditions.*

*The 385 disturbed surface soil samples were taken during a field campaign in February and March 2019 up to a depth of 5 cm. Since not all slopes were south-facing, transect directions were recorded in 8 directions of 45° angles each (e.g., N = 337.5 – 22.5°, NE = 22.5 – 67.5°). As the argan forest does not grow in a perfect grid pattern and the nearest tree in the sampling direction was not always in the exact direction needed, 90° differences between transect directions could not always be assured, but could vary some degrees to the left or right. For some trees less soil samples were taken due to the different tree densities and different tree architectures. For very dense, shrub-like trees it sometimes was not possible to sample the soil at the T2 sampling location because the soil was too well protected by the thorny, dense crown. Tree-tree distances on test sites with a high tree density could be very small, so that IT3 and IT4 sampling locations were the same on these test sites. Mean tree-tree distances on the test sites varied from 1.8 m to 17.5 m, with trees sometimes growing in tree groups and not being equally distributed on the test sites, so that some distances between trees could be much larger with a maximum of 50 m on one test site." (lines 124-144 in the revised manuscript).*

Study areas

I suggest rename this subsection as Study areas since you have studied three sites.

*Response: Thank you. We changed it.*

This subsection is too long. Please try to be more concise only highlighting the information is really relevant for this research.

*Response: We deleted figure 3 and table 1 and rewrote the text, thus shortening it. Each study area is now described in its own paragraph. It is now:*

*"The three study areas Ida-Outanane, Taroudant and Aït Baha are located in the western part of the Souss basin (Fig. 1, between 30° and 31° northern latitude and 9° and 7° western longitude). 30 test sites (one ha each) were chosen in these three environmentally differing study areas in order to cover varying altitudes, climate conditions and soil types (see Kirchhoff et al., 2019a).*

*Ida-Outanane is located on the southern foothills of the High Atlas close to Agadir and the Atlantic Ocean. Thus, its climate is more maritime with temperatures very rarely exceeding 30 °C and precipitation ranging from 230-260 mm (data for the suburbs of Agadir, 20 km away) (Díaz-Barradas et al., 2010; Saidi, 1995). Soils are mostly immature with Regosols, Leptosols and Fluvisols (Jones et al., 2013) covering Paleozoic, Mesozoic and Cenozoic rocks of the High Atlas (Hssaisoune et al., 2016). Traditional rainfed agriculture (mostly wheat cultivation) is practiced on three out of six test sites between the argan trees while the rest is under silvopastoral land use.*

*The study area of Taroudant also lies in the southern foothills of the High Atlas, but is situated further inland about 80 km from the coast. The climate is more continental with 220 mm annual precipitation and a mean annual temperature of 20 °C (Peter et al., 2014; Saidi, 1995). Eleven test sites are situated in the study area, seven on a loamy alluvial fan covering the Pliocene and Quaternary fluvial, fluvio-lacustrine and aeolian deposits of the Souss basin (Aït Hssaine and Bridgland, 2009; Chakir et al., 2014), the other four sites are located on the foothills of the High Atlas. The vegetation is mainly characterized by Argania spinosa as well as other shrubs and bushes such as Launaea arborescens, Ziziphus lotus, Acacia gummifera, Euphorbia spec. and Artemisia spec. (Ain-Lhout et al., 2016; Peter et al., 2014; Zunzunegui et al., 2017). However, a dynamic land use change has been taken place in the Souss basin for several decades, with traditional rainfed agriculture and argan trees being replaced by more profitable irrigated citrus plantations as well as greenhouses for banana and vegetable cultivation (d'Oleire-Oltmanns et al., 2012; Kirchhoff et al., 2019b; Peter et al., 2014).*

*The study area Aït Baha is located on the northern foothills of the Anti-Atlas Mountains. Precipitation ranges from 250-350 mm annually and the annual temperature averages 18.7 °C (Seif-Ennasr et al., 2016). The Anti-Atlas is mostly made up of Precambrian and Paleozoic rocks, which are covered by Fluvisols (Jones et al., 2013) as well as Regosols and Leptosols. Thirteen test sites are situated in this study area, with three on argan reforestation sites that often yield mixed results (Defaa et al., 2015). Silvopastoral land use dominates on most test sites with cereals being cultivated between the argan trees on ploughing terraces (on three test sites)."* *(lines 80-105 in the revised manuscript).*

Figure 2 should be placed at the bottom of this subsection. A new data frame on the corner including whole Morocco and where the areas are could enrich the figure.

*Response: Thank you for this suggestion. We moved the figure to the end of the section and improved the locations of the study areas. Regarding the small overview map, we would like to avoid making political statements regarding the status of the West Sahara, since official opinions in this matter vary between the home countries of the co-authors. We have therefore chosen the map extent so as to not to show the southern border of Morocco.*

Please use international units properly (e.g., 30ºC)

*Response: Thank you for pointing this out. We changed it throughout the text.*

In my opinion, Figure 3 and Table 1 are superfluous. They can be useful but they are not necessary for a research manuscript in which the capacity of summarize contents is crucial. I guess the journal provides the opportunity of using supplementary materials.

*Response: Thank you for your comment. We removed Figure 3 and Table 1.*

Lines 129-132: It forms part of your experimental design. In fact, perhaps you should write a new subsection called Experimental design in which you can add the figure 1, your sampling strategy, etc.

*Response: We added a subsection Experimental design according to your suggestions.*

Methods

I totally disagree with the name of this subsection as well as with its subdivision in extra and unnecessary sub-subsections (2.2.x. level). Please restructure your entire M&M section in logical and standard subsections, i.e., use only 2 and 2.x levels.

*Response: Thank you for the comments and solutions for restructuring the Material and methods section. As mentioned above, we restructured it the following way according to your suggestions. 2 Material and methods, 2.1 Study areas, 2.2 Experimental design (with figures for potential influences and soil sampling), 2.3 Soil analyses, 2.4 Tension-disc infiltrometer experiments, 2.5 Statistical analyses.*

Soil sampling and analyses should be your third subsection.

*Response: Thank you for this suggestion. We chose soil analyses as third subsection and explained soil sampling in the subsection Experimental design.*

Figure 4: The experimental design is well-thought. You have collected grosso modo 13 samples per tree and you have surveyed 30 trees. It is robust. Well-done!

*Response: Thank you.*

How many samples have you collected in each one of the three areas: 10 trees per tree areas plus inter-trees area?

*Response: We sampled 13 trees in the study area of Aït Baha, seven in Ida-Outanane and eleven in Taroudant. We added a sentence to show it in the manuscript: "One tree per test site was investigated, except on one test site where sampling was undertaken around two trees (31 trees in total), therefore sampling 13 trees in the study area of Aït Baha, seven trees in the study area Ida-Outanane and eleven trees in the study area of Taroudant." (lines 113-115 in the revised manuscript).*

Results

I am quite satisfied with this section. So, I have no comments that can be useful in this matter. The only thing that I suggest some modifications is regarding tables. They could be more

standard (no colours, etc.). From a visual point of view I do not like them. It is just an aesthetical suggestion.

Line 223: Tables should be placed below the paragraphs in which they are mentioned not above.

*Response: Thank you for pointing this out. We have moved the table and altered Table 3 and Table 4, so they do not contain coloured cells anymore. To differentiate values to the reader, we now used bold, italic or underlined text.*

Line 248: Why have you measured Kh only in 19 sites? It is just for curiosity. It does not mean the study is not rigorous.

*Response: Unfortunately, we were not able to measure Kh on all test sites during our field work in October/November 2019 because of time constraints. However, we think that with 19 out of 30 test sites we covered a reasonable part of the differences in our study areas and test sites.*

Discussion

In my opinion, the content of this section and the logical order of its writing as well as the most of the references used are appropriated. I have missed some pertinent references that could be quoted but I am not going to suggest any specific work. It is a task that corresponds exclusively to authors.

Line 352: There is no information about the 30 trees selected in any part of the text. If you have measured some variables about them it would be great if is shown in the text or as supplementary material.

*Response: We have added some information about the variation of trees to the subsection Experimental design: „Therefore, sampled trees were between 1 and 8 m high and varied from tall trees with round crowns to very dense shrub-like tree forms. Tree density varied from 3 to 292 trees ha$^{-1}$.“ (lines 117-118 in the revised manuscript). However, more research is needed on how or if the different tree architecture or degradation states influence the soil. We plan on investigating this issue in a further study but so far have not done so.*

Line 367: I have assessed some reforestations in northern Africa and I am very skeptical about their success. I think measures to prevent degradations are more effective.

*Response: We mostly agree with your assessment, we have, however, witnessed some (few) reforestation measures that were effective when protected and cared for long enough. This is especially important for a slow-growing tree like Argania spinosa. It is still very important that the degradation by overgrazing is prevented.*

Line 370: Do you 100% agree with this sentence? How can we change micro-climates? Please be more "prudent" in this kind of sentences.

*Response: Thank you for pointing this out to us. We deleted this subclause.*

Conclusion

I agree with the content expressed in this section but I think it should not be a simple summary of your results as the first part is. Anyway, it is well-written.

*Response: Thank you for this comment. We re-wrote the conclusion so it is not a summary of the results. It is now: "In this study, we were able to show that spatial patterns of tree influence on soil properties exist around argan trees. They are most pronounced due to, e.g., litter translocation to the east and downslope of the tree and due to the effect of midday shade to the north of the tree on gravimetric soil water content. The decrease of the studied soil parameter values with increasing distance from the tree trunk suggests that a fertile island structure is concentrated solely around the trunk while the rest of the tree area is still protected by the canopy, as visible in the infiltration properties, but is more similar to the intertree area.*

*Reforestation measures should aim to plant young sprouts close to the trees, ideally in northern or eastern directions to take advantage of the shade (higher soil water content) and the higher content of soil organic carbon and nitrogen. Since not all argan trees are similar in size, tree architecture and genetic variety, more research is needed on how these factors influence the soil under argan trees." (lines 373-381 in the revised manuscript).*

Referee 2

**GENERAL COMMENTS**

The topic of the work is current in terms of maintaining the fertility of the soil under arid conditions. It is an interesting study to evaluate spatial patterns of argan-tree influence. In general, this manuscript is acceptable, but the following issuers are still needed to consider.

**SPECIFIC COMMENTS**

**Title**

To my knowledge the use of the term soil quality is not appropriate in this study. There is a wide literature assessing soil quality, although some authors use a few parameters it is preferable to combine them and create a soil quality index (SQI).

*Response: Thank you for this comment. We agree that the term soil quality is misleading and have changed the title accordingly to: „Spatial distribution of argan-tree influence on soil properties in South Morocco". We have also changed this issue in the text.*

**Abstract**

Line 8-12: More generally, I suggest focusing the paragraph on the conducted study.

*Response: Thank you, we added „…however, it is unknown if the trees influence the soil of the intertree areas." (line 10 in the revised manuscript).*

Line 18-23: Some data obtained should be included

*Response: We added SOC-data to show the decrease from tree trunk along tree drip line to intertree area. It is now: "We found that the tree influence on its surrounding intertree area is limited, with e.g., SOC- and TN-content decreasing significantly from tree trunk (SOC: 4.4 %, TN: 0.3 %) to tree drip line (SOC: 2.0 %, TN: 0.2 %). However, intertree areas near the tree drip line (SOC: 1.3 %, TN: 0.2 %) differed significantly from intertree areas between two trees (SOC: 1.0 %, TN: 0.1 %), yet only with a small effect." (lines 17-20 in the revised manuscript).*

Line 18: I suggest modifying the nomenclature: SOC to refer soil organic carbon and TN to refer total nitrogen. This should be modified in the whole text.

*Response: Thank you for this comment. We have changed the nomenclature to SOC and TN throughout the text.*

**Introduction**

I recommend reducing the length of the text while preserving the informative value in terms of brevity.

*Response: Thank you for this recommendation. We deleted some sentences or merged them with others to reduce the length of the text.*

I suggest moving Figure 1 to Material and Method section.

*Response: Thank you. We moved it to the Material and methods section.*

The authors should highlight the meaning or the purpose of this study in the introduction part. What kind of gap could you fill by doing this study?

*Response: Thank you for this comment. In the last paragraph of the introduction we write about the research gap about spatial patterns of influence around argan trees. So far, only Zinke (1962) has written about influence patterns of single trees, but his research focused on pine forests in California and not on dryland forests. This last paragraph is based solely about the purpose/gap to be filled by this study:*

*"This hypothetical spatial pattern of influence of the tree on the soil has only been researched for pine trees (Zinke, 1962) but not at all for argan trees or dryland forests whereas the differences between tree or shrub vegetation and their corresponding intertree/intershrub areas have been investigated before, especially in 'fertile island'-research (e.g., Belsky et al., 1993; Boettcher and Kalisz, 1990; de Boever et al., 2015; Pérez, 2019; Qu et al., 2018). In South Morocco, where the geomorphologic processes are highly dynamic (Aït Hssaine, 2002; Kirchhoff et al., 2019b; Marzen et al., 2020; Peter et al., 2014), it is likely that litter and soil particles are dislocated to the intertree areas. The knowledge about this possible dislocation and improvement of soil parameter values in the intertree areas could enable a better regrowth in these areas (Boulmane et al., 2017; Defaa et al., 2015) or show the need for rehabilitation by limiting degradation factors like overgrazing.*

*The aim of this study is therefore to analyse the spatial distribution of the influences an individual argan tree has on soil properties of the intertree areas. For this purpose, we define*

*- "tree area" as the area covered by canopy (within the tree drip line),*

*- "intertree area" as all area not covered by canopy (i.e., between tree areas)." (lines 66-77 in the revised manuscript).*

**Material and Methods**

The Material and Methods section is poorly organized and very confusing.

*Response: We have reorganised the Material and methods section. It is now: 2.1 Study area, 2.2 Experimental design (where we also moved Fig. 1 from the introduction), 2.3 Soil analyses, 2.4 Tension-disc infiltrometer experiments, 2.5 Statistical analyses.*

I suggest major re-write of the study area section.

*Response: Thank you for the suggestion. We re-wrote the section, thus reducing the length of the section. Each study area is now described in its own paragraph. It is now:*

[revised manuscript text omitted]

Figure 3 only provide precipitation at Aït Baha. Could you provide information of the other study areas?

*Response: Because of a comment of Reviewer 1, we deleted Figure 3.*

Line 104: Pleas clarify "ca."

*Response: Because of the re-write of the section „Study area", we deleted this part.*

Line 104: "Figure 3 shows that in recent years the annual precipitation of this study area has decreased to ca. 220 mm, possibly a sign of higher aridity due to climate change". This a very vague statement. In order to make this statement, a greater period of years is necessary, and Figure 3 only 15 years were included.

*Response: Thank you for this comment. We deleted this part due to the re-write of the section.*

Line 156: Please describe how 1-2 mm aggregates were obtained.

*Response: The aggregates were sieved for the size 1-2 mm. We added this to the text.*

Under which soil conditions were the infiltration measurements carried out? Were they homogeneous?

*Response: We measured the unsaturated hydraulic conductivities in October/November 2019, when the soils were very dry. We took care to always measure the sampling locations T2 and IT3 at the same time for each test site, so we could compare the two sampling locations. The soil texture classes of T2 and IT3 were the same for 16 out of 19 test sites. We also added some information in the text: "At the time of the measurements in October/November 2019 soils were very dry in all three study areas (soil water content at the measurement points: 0.1 – 0.6 %). Soil texture classes were the same for T2 and IT3 sampling locations on 16 out of 19 test sites." (lines 182-185 in the revised manuscript).*

**Results**

Line 202-205: delete, it is described in table 2

*Response: Thank you. We deleted it.*

This section is well structured and well written. Illustrations used in the text are very useful and high quality.

*Response: Thank you.*

**Discussion**

Line 301-302: Reiterate the objective of the work is not necessary.

*Response: Thank you. We have deleted the first sentence.*

In some lines it is recommended to include current references (eg. 309 or 344).

*Response: We have added some current references for these lines, namely De Boever et al. 2015, and Yang et al. 2020 (lines 308 & 346 in the revised manuscript).*

De Boever, M., Gabriels, D., Ouessar, M., and Cornelis, W.: Influence of scattered Acacia trees on soil nutrient levels in arid Tunisia, Journal of Arid Environments, 122, 161–168, doi:10.1016/j.jaridenv.2015.07.006, 2015.

Yang, C., Geng, Y., Fu, X. Z., Coulter, J. A., and Chai, Q.: The Effects of Wind Erosion Depending on Cropping System and Tillage Method in a Semi-Arid Region, Agronomy, 10, 732, doi:10.3390/agronomy10050732, 2020.

Line 322: "The medium to large effect for the unsaturated hydraulic conductivities could be explained by the higher porosity due to a higher content of organic" However porosity data is not available in the study. In this sense soil physical properties such as textural class or bulk density on microsite locations canopy and outside canopy for Argan trees are particularly relevant especially in an overgrazed environment. In addition, unsaturated hydraulic conductivity highly depends on soil's particle size distribution.

*Response: Unfortunately, we did not measure soil porosity or bulk density in this study. We therefore changed the sentence: „De Boever et al. (2014) found that organic material correlated well with soil porosity and bulk density thus possibly explaining the medium to large effect for the unsaturated hydraulic conductivities between T2 and IT3 sampling locations." (lines 320-322 in the revised manuscript). As mentioned above for the Material and methods, we added information if there were texture class differences between T2 and IT3 sampling locations.*

Line 324-325: "In a previous study, we found higher erosion rates as well as lower infiltration rates in the intertree areas". Please provide data erosion rates as well as infiltration rates.

*Response: We have added the suspended sediment concentrations as well as infiltration rates from the previous study and replaced erosion rates with suspended sediment concentration rates. It is now: "In a previous study, we found higher average suspended sediment concentrations (4.42 g $L^{-1}$ compared to 2.18 g $L^{-1}$ under argan trees) as well as lower average infiltration rates in the intertree areas (229.56 mm $h^{-1}$ compared to 452.57 mm $h^{-1}$ under argan trees) (Kirchhoff et al., 2019a)." (lines 324-326 in the revised manuscript).*

Line 343: "large aggregates" Macroaggregates seems more appropriate.

*Response: We have changed it to macroaggregates.*

Line 352: "The type of tree (architecture, size, genetic variety) could be a possible explanation for the missing significance of the directions." However, there is no information in the text about type of tree selected (eg. Crown diameter or tree age) and it seems to be a relevant aspect in the conclusions. Has any pattern been followed to choose the analysed trees? If you have measured some parameters related to tree typology, please add them.

*Response: We have added some information about the variation of trees to the subsection Experimental design: „Therefore, sampled trees were between 1 and 8 m high and varied from tall trees with round crowns to very dense shrub-like tree forms. Tree density varied from 3 to 292 trees $ha^{-1}$." (lines 117-118 in the revised manuscript). However, more research is needed on how or if the different tree architecture or degradation states influence the soil. We plan on investigating this issue in a further study but so far have not done so.*

Line 361: "Although the soil quality decreases from T1 to T2, the T2". A soil quality index has not been developed to support this statement.

*Response: Thank you for this comment. We changed soil quality to „most of the measured soil parameter values" (line 362 in the revised manuscript).*

Line 364: (Qu et al., 2018) reference missing.

*Response: Thank you for pointing this out. We have added the reference.*

*Qu, L., Wang, Z., Huang, Y., Zhang, Y., Song, C., and Ma, K.: Effects of plant coverage on shrub fertile islands in the Upper Minjiang River Valley, Science China Life Sciences, 61, 340-347, doi: 10.1007/s11427-017-9144-9, 2018.*

Relevant changes to the manuscript

General

- *Changed "$C_{org}$", "N" and "&" to "SOC", "TN" and "and"*
- *Reordered references in the text alphabetically*
- *Changed soil moisture to soil water content*

Title

- *Change of title to "Spatial distribution of argan-tree influence on soil properties in South Morocco"*

Abstract

- *Added SOC and TN-data*
- *Change of some sentences → see referee responses (pages 1, 2, 8, 9)*

Introduction

- *Added literature → see referee responses (page 3)*
- *Shortened introduction*
- *Moved Figure 1 → now Fig. 2 (potential influences) to Experimental design*
- *Changed some sentences → see referee responses (pages 3, 9, 10)*

Material and methods

- *Rewrote 2.1 Study area with removal of (old) Fig. 3 and (old) Tab. 1; added mini-map and rectangles showing study areas to Fig. 2 → now Fig. 1 (Study area) → see referee responses (pages 4, 5, 6, 10, 11, 12)*
- *Changed structure of Material and methods → 2.1 Study area, 2.2 Experimental design, 2.3 Soil analyses, 2.4 Tension-disc infiltrometer experiments, 2.5 Statistical analyses*
- *Added Fig. 1 → now Fig. 2 (Potential influences…) to Experimental design*
- *Changed colour scheme in Fig. 4 → now Fig. 3 (sampling design)*

- *Expanded explanation about Experimental design → see referee responses (pages 4, 6, 7, 11, 13)*
- *Added some information about measurement time to section 2.4 Tension-disc infiltrometer experiments → see referee responses (page 12)*

Results

- *Corrected values for soil water content*
- *Changed Table 3 → now Table 2 (effect sizes) to not contain coloured cells anymore*
- *Changed colour scheme in Fig. 5, 6, 7, 8, 9 → now Fig. 4, 5, 6, 7, 8*

Discussion

- *Added literature → see referee responses (pages 3, 12, 13)*
- *Changed, added or removed some sentences → see referee responses (pages 7, 12, 13, 14)*
- *Added data from previous research → see referee responses (page 13)*

Conclusion

- *Rewrote conclusion → see referee responses (page 8)*